# Analysis of Nephrolithiasis Treatment in Highest Reference Hospital—Occurrence of Acromegaly in the Study Group

**DOI:** 10.3390/jcm12123879

**Published:** 2023-06-06

**Authors:** Tomasz Ząbkowski, Adam Daniel Durma, Agnieszka Grabińska, Łukasz Michalczyk, Marek Saracyn

**Affiliations:** 1Department of Urology, Military Institute of Medicine—National Research Institute, 02-637 Warsaw, Poland; 2Department of Endocrinology and Radioisotope Therapy, Military Institute of Medicine—National Research Institute, 02-637 Warsaw, Poland; 3Urology Department, District Hospital, 26-900 Kozienice, Poland

**Keywords:** acromegaly, urolithiasis, URSL, RIRS, PCNL, ESWL

## Abstract

Background: Urolithiasis is one of the most common diseases of the urinary system, the incidence of which is assumed to be up to 100,000 cases per million (10% of the population). The cause of it is dysregulation of renal urine excretion. Acromegaly is a very rare endocrine disorder that causes a somatotropic pituitary adenoma producing higher amounts of growth hormone. It occurs approximately in 80 cases per million (about 0.008% of the population). One of the acromegaly complications may be urolithiasis. Methods: Clinical and laboratory results of 2289 patients hospitalized for nephrolithiasis in the highest reference hospital were retrospectively analyzed, distinguishing a subgroup of patients with acromegaly. Statistical analysis was performed to compare the prevalence of the disease in the analyzed subgroup with the epidemiological results available in up-to-date literature. Results: The distribution of nephrolithiasis treatment was definitely in favor of non-invasive and minimally invasive treatment. The methods used were as follows: ESWL (61.82%), USRL (30.62%), RIRS (4.15%), PCNL (3.1%), and pyelolithotomy (0.31%). Such a distribution limited the potential complications of the procedures while maintaining the high effectiveness of the treatment. Among two thousand two hundred and eighty-nine patients with urolithiasis, two were diagnosed with acromegaly before the nephrological and urological treatment, and seven were diagnosed de novo. Patients with acromegaly required a higher percentage of open surgeries (including nephrectomy) and also had a higher rate of kidney stones recurrence. The concentration of IGF-1 in patients with newly diagnosed acromegaly was similar to those treated with somatostatin analogs (SSA) due to incomplete transsphenoidal pituitary surgery. Conclusions: In the population of patients with urolithiasis requiring hospitalization and interventional treatment compared to the general population, the prevalence of acromegaly was almost 50-fold higher (*p* = 0.025). Acromegaly itself increases the risk of urolithiasis.

## 1. Introduction

Urolithiasis (kidney stones) is a condition of the presence of urinary tract deposits. They result from the precipitation of chemicals contained in the urine when their concentration exceeds the solubility threshold. The deposits are most often composed of calcium oxalate crystals (rarely of calcium phosphates, urates, cysteine, or struvite). In the largest percentage, these changes are located in the renal calyces or renal pelvis, although they can move to the ureter, causing obstructive uropathy and, consequently, hydronephrosis, bacterial infection, and urosepsis [1]. Such changes may remain asymptomatic for many years or cause severe pain and kidney dysfunction. One of the most dangerous complications may be pyonephrosis which can lead to organ loss or even death [2,3]. Urolithiasis is characterized by a relatively high prevalence of 4.4–10.1% depending on the population [4,5], as well as a high risk of recurrence estimated up to 50% over 10 years [6]. Recurrence of the disease depends on the supply of fluids and diet, as well as genetic predisposition. Due to the fact that diet has the greatest influence on the development of kidney stones, proper diet, and hydration are crucial in the treatment of the disease [7,8,9]. The divergences in diet patterns can also be a possible reason for the variability in the prevalence of the disease in different populations [10,11]. Modern methods of treating kidney stones allow for non-surgical and often even non-invasive removal of deposits, ensuring both pain relief and reducing the risk of serious complications. It should be emphasized that due to the high risk of recurrence, patients treated for urolithiasis often need multiple and various types of procedures to control the disease [6].

One of the most frequently performed procedures, due to its availability, low invasiveness, and low risk of complications, is the crushing of urinary deposits using extracorporeal shock-wave lithotripsy (ESWL). For the procedure, patients with kidney stones less than 20 mm in diameter are qualified. Most often, the procedure is performed under short analgosedation, with the use of a lithotripter placed in the vicinity of the kidney projection. After locating the kidney stones using ultrasonography, the ultrasound waves focused on the deposits are administered to crush deposits. The deposits are divided into small fragments that are spontaneously excreted with the urine [12,13,14]. An important aspect of the method is the ability to conduct the entire procedure in outpatient clinic conditions.

URSL (UreteroRenoScopic Lithotripsy) is a method based on crushing ureteral stones using a mechanic lithotripter or laser fiber after inserting a rigid ureteroscope into the ureter [15,16]. The operator decides on the choice of the method and place of its application. Due to the operating conditions (stenosis, swelling of the ureter, size of the stone, or calculi), the URSL can be a two-stage procedure—in the first stage, insertion of a double-J catheter is performed, followed by the action of the kidney stone.

RIRS (retrograde intrarenal surgery) is a method that is performed with a flexible ureterorenoscope, where kidney stones are crushed in the renal calyces and renal pelvis by the use of a laser beam [17,18]. The method is often introduced when ESWL fails.

PCNL (PerCuteneous NephfroLithotripsy) is a method of crushing large deposits (over 20 mm) in the pelvicalyceal system—for example, cast urolithiasis or numerous deposits. The technique of the procedure consists in creating a renal fistula and inserting a stiff endoscope (nephroscope) into the kidney, which crushes the deposits that are removed from the kidney. The kidney puncture is performed under ultrasound and X-ray guidance [19,20,21]. In the event of massive bleeding from the renal parenchyma or technical problems during the operation, the operator may decide to end PCNL and convert the procedure to classic surgery. It is notable that, in the case of massive bleeding, kidney loss may occur.

Pyelolithotomy is a classic (open) surgical method currently performed mainly in the case of severe cast nephrolithiasis when other methods are impossible or contraindicated [22,23]. In the largest percentage of cases, this method is used in the case of struvite urolithiasis, associated with the formation of cast stones and filling most or all of the calyces and renal pelvis [24,25]. The choice of the surgical method depends on the conditions of the patient (urine tract developmental defects and abnormalities, patient comorbidities) and the experience of the surgical (urological) team. Patients with frequent recurrence of urolithiasis should be diagnosed with metabolic and genetic disorders.

The appearance of nephrolithiasis, as well as its severity, may be the result of numerous endocrine disorders. The endocrinopathies leading to urolithiasis are ones resulting in hypercalcemia or disrupted excretion of urine calcium or phosphates [26,27]. The most common endocrine cause of hypercalcemia is primary hyperparathyroidism. This condition caused by parathyroid adenoma or adenomas leads to excessive secretion of parathyroid hormone, causing increased calcium absorption in the intestines and increased calcium reabsorption in the kidneys, as well as hyperphosphaturia. The most common non-endocrine cause of urolithiasis in the population is malignant neoplastic diseases (breast, kidney, lung, gastric or intestinal cancer) [28]. In the case of malignant tumors, hypercalcemia is caused by excessive synthesis of a peptide structurally related to parathyroid hormone (PTHrP), production of osteolytic cytokines, and excessive production of 1,25-(OH)_2_D_3_ due to cytokines which are secreted by tumor cells. It is estimated up to 30% of cancer patients. Hypercalcemia is seen especially in patients in the later stages of the disease. The two above-described causes account for almost 90% of all cases of elevated blood and urine calcium levels in the population. However, it should be remembered that hypercalcemia may also be caused by granulomatous disease, where deregulation of the 1,25-(OH)_2_D_3_ production by activated macrophages causes an increase in active vitamin D concentration and its systemic effect [29]. It is estimated that even 10% of patients with sarcoidosis (or another granulomatous disease) may present hypercalcemia in laboratory tests.

Acromegaly is a rare endocrine disease, with a prevalence of 2.8–13.7 per 100,000 patients (about 0.008% of the population) [30]. It is mainly caused by pituitary adenoma, which causes excessive and uncontrolled secretion of growth hormone (GH); however, the presence of neuroendocrine neoplasm (NEN) in any location, which secretes GH or growth hormone-releasing hormone (GHRH) is also possible [31]. The excessive secretion of GH leads to increased production of insulin-like growth factor 1 (IGF-1) and stimulation of tissues and organ growth, as well as stimulation of certain metabolic pathways.

Acromegalia can be undetected for a long period of time due to its unspecific symptoms and signs, which develop after years of hormonal stimulation to human body tissues [32]. Previous observations indicate that the time from disease onset to diagnosis and initiation of treatment can even take up to 10 to 11 years. Furthermore, studies on the Caucasian population (which are also the subject of analysis in this manuscript) have shown that the average age of disease diagnosis is 47.3 ± 14.1 years. There was also a predominance of females (60% of cases compared to 40% of males) [33].

The primary screening test for the diagnosis of acromegaly is the measurement of IGF-1 levels. If the concentration of this parameter exceeds the normal range, an oral glucose tolerance test with 75 g of glucose (OGTT) is performed [34]. The diagnostic threshold for acromegaly is a GH concentration of 1.0 μg/L (ng/mL), or 0.4 μg/L in the case of ultrasensitive GH assay [35,36]. Since the majority of cases are caused by somatotropin-secreting pituitary adenomas, the primary treatment is surgery (transsphenoidal or transcranial approach in the case of large macroadenomas) [37]. In cases of inoperable tumors or incomplete surgeries, first-generation somatostatin analogs (octreotide, lanreotide), second-generation SSA (pasireotide), or GH receptor agonists (pegvisomant) are subsequently used [38,39,40,41,42].

Despite causing organ growth, the increased concentration of GH and IGF-1 stimulates many metabolic pathways. One of them can lead to hypercalcemia and the formation of deposits in the urinary system [37,43]. Acromegalia can be undetected for a long period of time due to its unspecific symptoms and signs, which develop after years of IGF-1 stimulation to human body tissues, causing complications of the disease multiple and advanced. Thus the nephrolithiasis diagnosed in acromegalic patients is very often demanding urological intervention or surgery [44].

The aim of the manuscript was to analyze the correlation between acromegaly and nephrolithiasis in the group of patients who required hospitalization in the highest reference center (and urological intervention).

## 2. Materials and Methods

The data of patients hospitalized due to nephrolithiasis in the Urology Department of the Military Institute of Medicine—National Research Institute in the years 2020–2022 were retrospectively analyzed. It should be mentioned that the Department is the highest reference center in the country, and the years 2020–2021 are considered the peak of the SARS-CoV-2 pandemic, which resulted in a partial limitation of access to medical services (diagnostic and therapeutic procedures). Laboratory tests were performed in the Department of Medical Diagnostics Military Institute of Medicine—National Research Institute. Statistical analysis was performed using the IBM SPSS Statistics package, Version 25.0., Armonk, NY, USA: IBM Corp. (2021), using a 2-sample *t*-test with unequal variances and one-tailed hypothesis assumption. The level of significance was assumed at *p*-value < 0.05.

## 3. Results

In the years 2020–2022, the total number of patients hospitalized for nephrolithiasis in the Urology Department was 2289. The main treatment procedure performed was ESWL (61.82%), followed by the USRL (30.62%) and RIRS (4.15%). The lowest percentage of patients had performed procedures disrupting skin continuity—PCNL (3.1%) and pyelolithotomy (0.31%). The quantitative distribution of procedures performed in the analyzed period is presented in Table 1 and Figure 1.

During hospitalization, in nine patients (9/2289—0.39%), due to typical phenotypic changes characteristic of acromegaly (changed facial morphology, widening of interdental spaces, inadequate enlargement of hands and feet), endocrinological consultation was referred. Among the group of nine patients, two had already been diagnosed with acromegaly and had been treated surgically in the past (incomplete transsphenoidal removal of the pituitary adenoma). These two patients had been treated pharmacologically with the use of long-acting somatostatin analogs.

Among the remaining seven patients, acromegaly was preliminarily confirmed on the base of the screening test—concentration of insulin-like growth factor 1 (IGF-1)—which was above the reference range. The acromegaly was later confirmed in the Endocrinology Department of another reference center. The detailed results of the analyzed subgroup are presented in Table 2.

It is notable that the results of IGF-1 in patients with pharmacological acromegaly treatment were not significantly different than in ones with newly diagnosed acromegaly. Moreover, in the analyzed subgroup of nine patients with acromegaly, three (33%) of them underwent nephrectomy in the past due to cast kidney stones complicated by pyonephrosis. In the remaining (the only) kidney, renal or ureteral calculi recurred regularly. In the analyzed period of observation, seven patients (78%) required multiple interventions due to recurrent pain associated with the presence of deposits in the urinary tract. All analyzed patients with acromegaly and kidney stones are currently maintaining their kidney function thanks to regular ESWL, URSL, and RIRS procedures or stenting of the ureters with double J-catheters. Figure 2 presents procedures made in the subgroup of acromegaly patients.

In our analysis, we also compared results obtained retrospectively with mean populational data available in the most recent literature. Comparing those data, we noticed that the presence of urolithiasis (which demanded urological intervention) had been increasing the risk of acromegaly diagnosis almost 50 times (*p* = 0.025).

## 4. Discussion

The analysis of the distribution of procedures performed in our center (highest reference one) shows that the development of urology treatment techniques and the direction of therapeutic trends are aimed toward non-invasive methods. Nearly two-thirds of urolithiasis treatment procedures were performed using the non-invasive method of ultrasonic stone crushing from an external source. Such a high percentage of procedures is achieved due to the simplicity of the procedure, as well as the possibility of performing it on an outpatient basis (one-day clinic). This possibility reduces probable complications related to tissue intervention but also effectively affects the economic aspect of treatment (a smaller amount of necessary medical documentation, reduced involvement of medical staff, reduced cost of the procedure, and possibility to perform a higher number of procedures in the same period of time). More than one-third of urolithiasis treatments are endoscopic procedures, which advantage is low invasiveness and the possibility of direct action on deposits. The disadvantage of such solutions is the need for qualified staff and the availability of hospital facilities. However, thanks to the direct action on deposits, the percentage of effective treatment can be higher. The smallest percentage of procedures are procedures involving the violation of the skin layers, requiring both full anesthesiologic protection, as well as the highest number of possible complications. However, it should be remembered that although these methods are reserved for the most severe cases of urolithiasis, a failure of less invasive methods may result in the need to perform invasive procedures.

The analysis of the presented group of patients with urolithiasis showed that in this group, the incidence of acromegaly was 0.39%, which was almost 50-times higher than the estimated prevalence of acromegaly in the general population (approximately an average of 0.008%). A greater percentage of acromegalic patients (78% vs. 50%) required re-procedures due to recurrent urolithiasis—even ones with already introduced pharmacological treatment (however, they were not optimally guided) which proves the persistent, negative effect of GH and IGF-1 on tissues and metabolic pathways, and calcium/phosphate metabolism. The other fact is that acromegaly was more often diagnosed in women (7 vs. 2). The most important reason for these differences may arise from more visible and significant changes in morphological presence in women (the disease quicker disrupts the esthetics of the female face). Furthermore, sex hormones can also influence renal 1-hydroxylase by increasing its activity, which is followed by an increase in the concentration of active vitamin D [45].

Over 40 years ago, the connection between acromegaly and nephrolithiasis was noticed, and the cause of the aggravation of kidney stones was identified [46,47]. Excessive concentrations of GH and IGF-1 in patients with acromegaly stimulate the synthesis of 1,25-dihydroxyvitamin D (1,25(OH)_2_D), which deregulates the calcium and phosphate metabolism leading to hypercalcemia [48]. Parathyroid hormone (PTH), fibroblast growth factor 23 (FGF23), calcium, and phosphate are the main regulators of renal 1-hydroxylase (CYP27B1)—the enzyme that converts 25-OHD to the active form 1,25(OH)_2_D. The main location of the process is the kidneys, although a certain percentage of the process occurs in peripheral tissues such as epithelial cells, immune system cells, and parathyroid glands [49].

The regulation of extrarenal 1-hydroxylase differs from the process ongoing in the kidneys and also involves the action of many cytokines such as tumor necrosis factor (TNFα), interferon gamma (IFNγ), or interleukins (IL-2, IL-4, IL-15) [50,51] Both GH and IGF-1, through a mechanism that has not been clearly confirmed and remains unclear, affect the stimulation of 1α-hydroxylase, which results in an increase in the concentration of active vitamin D in the plasma [51]. Hypercalcaemia may occur as a consequence of this inadequate increase. This mechanism is based on: increased intestinal calcium absorption, increased calcium reabsorption in the distal renal tubules, and inhibition of phosphate reabsorption, as well as activation of osteoclasts and increased bone demineralization [52,53]. Growth hormone and IGF-1 also have the potential to stimulate osteoclasts in a mechanism other than the activation of vitamin D. Chihara et al., in the studies on the effects of GH and IGF-1 on osteoblasts and osteoclasts, showed that GH works via direct as well as indirect stimulation of cell maturation precursor osteoclasts and through indirect activation of mature osteoclasts by cytokines and paracrine factors secreted by tissue stromal cells [53]. Ueland et al. confirmed both direct and indirect effects of GH and IGF-1 on osteogenic and osteogenic cells, mainly by affecting the balance between osteoprotegerin (OPG) and nuclear factor kappa β activator receptor (RANKL) [54]. The cumulative effect of these mechanisms leads to an increase in blood calcium concentration, total 24-h urinary calcium excretion, and ultimately, to the deposition of calcium salts in the renal pelvis, leading to the formation or build-up of deposits.

It should also be emphasized that an additional aspect increasing the risk of urolithiasis in patients with acromegaly may be the coexistence of hyperparathyroidism, e.g., in the course of MEN1 syndrome. Mutation of the MEN1 gene encoding the menin protein causes the loss of the normal function of the gene. Lack of menin increases the risk of cancer in the endocrine glands. The most common ones are the parathyroid glands—primary hyperparathyroidism—which occurs in about 95% of cases of the syndrome, and the pituitary gland—lactotropic, somatotropic, or non-functioning adenomas—which occur with a frequency of 30–40%. However, it is worth adding that if the MEN1 mutation is diagnosed and a pituitary adenoma occurs, only 5–25% of patients will develop acromegaly symptoms [55].

Very similar observations regarding the correlation between acromegaly and nephrolithiasis have been previously reported, consistent with the findings obtained in our study. Thai et al. presented a case of a 73-year-old Caucasian male whose recurrent, severe nephrolithiasis led to the diagnosis of acromegaly more than 20 years after the initial symptoms of kidney stones were detected [56]. Another study by van der Valk et al. demonstrated the case of a 53-year-old male with recurrent kidney stones containing calcium oxalate. This was also one of the first symptoms of acromegaly, which subsequently resolved with proper treatment—somatostatin analogs followed by surgery [57].

It should be noted that there is a lack of a significant number of well-designed prospective studies evaluating the correlation between acromegaly and kidney stones in available medical databases. Due to the high prevalence of kidney stones in the population, which is steadily increasing and predicted to reach up to 30% by 2050 [58,59], and considering the relative rarity of acromegaly in the population, a prospective study (particularly a blinded or randomized one) could be cost-ineffective regarding the expected benefits [60,61]. Nevertheless, if performed, especially with an extended analysis of vitamin D metabolites, it could give reliable data and explain ambiguities and uncertainties of the correlation between acromegaly and nephrolithiasis.

In summary, most recent studies and observation states that hypercalcemia in acromegaly results from the overproduction of GH and IGF-1, and furthermore, the activation of numerous mechanisms, which the dominant one seems to be an increase of the 1,25-dihydroxyvitamin D concentration. This process and activation of receptors on cells of the gastrointestinal tract, kidneys, and bones lead to elevated serum calcium concentration, followed by the formation of deposits in the urinary tract. Thus clinicians should pay attention to possible clinical features of acromegaly in order to detect the disease early and prevent its complication. In addition, in patients with acromegaly, observation of clinical features that may indicate nephrolithiasis should lead to the extension of diagnostics.

### Strengths and Weaknesses of the Study

The strengths of the work are the large number of analyzed patients, especially taking into account the time of the SARS-CoV-2 pandemic, which significantly limited access to medical services.

The weaknesses of the study are its retrospective nature and lack of data in the acromegalic subgroup—especially further endocrinological treatment of the patients.

## 5. Conclusions

Due to the increased risk of nephrolithiasis in acromegaly, it is important for patients to regularly check the condition of their kidneys and undergo periodic examinations in order to detect possible complications of urolithiasis early.

Urolithiasis may occur in patients with acromegaly at a much higher frequency than in the general population and may be one of the first systemic complications of the disease.

The frequency of recurrence of urolithiasis as well as its complications in the form of ureteral stenosis, hydronephrosis, or pyonephrosis, is higher in patients with acromegaly than in the general population.

Thanks to modern endoscopic methods of treatment of kidney stones and continuous improvement of double-J catheters, it is possible also to improve the quality of life of patients with acromegaly and kidney stones.

## Figures and Tables

**Figure 1 jcm-12-03879-f001:**
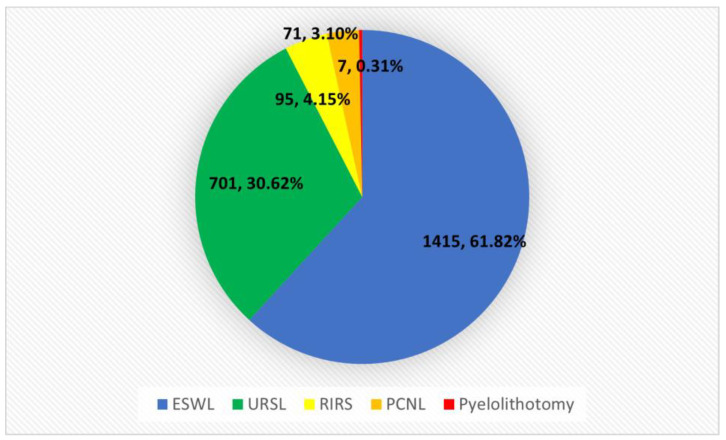
Diagram showing the percentage of kidney stones procedures performed in the years 2020–2022. Colors in the figure: Blue—extracorporeal shock-wave lithotripsy (ESWL); Green—UreteroRenoScopic Lithotripsy (URSL); Yellow—retrograde intrarenal surgery (RIRS); Orange—PerCuteneous NephfroLithotripsy (PCNL); Red—Pyelolithotomy.

**Figure 2 jcm-12-03879-f002:**
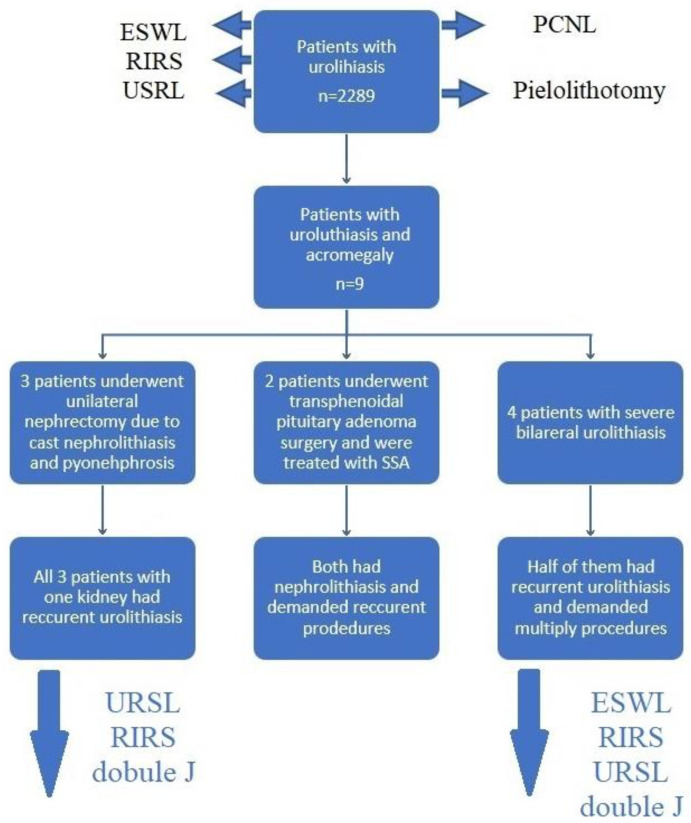
Procedures performed in acromegalic patients.

**Table 1 jcm-12-03879-t001:** Schedule of kidney stone treatment procedures in 2020–2022.

Year/Method	URSL	RIRS	PCNL	Pyelolithotomy	ESWL	Summary
2020	164	0	12	1	346	523
2021	235	27	23	1	495	781
2022	302	68	36	5	574	985
Summary	701	95	71	7	1415	2289

**Table 2 jcm-12-03879-t002:** Data analysis of a subgroup of patients with acromegaly.

Age	Mean	63.33
	SD	8.06
Gender	Female	7
	Male	2
Acromegaly diagnosis	Before nephrolithiasis	2
	After nephrolithiasis	7
Ca [N: 8.6–10.2 mg/dL]	Mean	9.97
	SD	0.62
Crea [N: 0.5–0.9 mg/dL]	Mean	1.06
	SD	0.36
IGF-1 (Total group *n* = 9) [ng/mL] *	Mean	370.89
	SD	101.32
IGF-1 (de novo group *n* = 7) [ng/mL] *	Mean	386.57
	SD	109.54

SD—standard deviation; IGF-1—insulin-like growth factor 1; Ca—calcium; Crea—creatinine; * Reference ranges for IFG-1 are gender- and age-dependent, nevertheless in every patient in the study, concentrations were above the upper normal limit.

## Data Availability

The datasets used and/or analyzed during the current study are available from the corresponding author upon reasonable request.

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
