# Peer review of "Analysis of Nephrolithiasis Treatment in Highest Reference Hospital—Occurrence of Acromegaly in the Study Group"

_jcm, 2023, doi:10.3390/jcm12123879_

Round 1
Reviewer 1 Report
There are too many general information about urolithiasis in the introduction, and acromegaly, the subject of this paper, is mentioned too briefly. This is a not textbook.
Authors used clinical and laboratory results of 2289 patients hospitalized 15 for nephrolithiasis in the highest reference hospital. And 9 were diagnosed with acromegaly. Based on this data alone, the authors concluded that “In the population of patients with urolithiasis compared to the general population, the prevalence of acromegaly was almost 50-fold higher.”
In my opinion, nephrolithiasis patients in highest reference hospital do not represent the whole patients. This is a serious selection bias. Basically, I think this study design is wrong. Therefore, the results of the study are also unreliable.
Author Response
Dear Reviewer,
First of all, we would kindly like to thank you for the review. We have corrected the manuscript according to all your valuable comments and suggestions. We hope the corrected manuscript will meet your all expectations. Below, we have attached the answers for all of your comments.
Comments and Suggestions for Authors
There are too many general information about urolithiasis in the introduction, and acromegaly, the subject of this paper, is mentioned too briefly. This is a not textbook.
The paragraph regarding acromegaly was updated. Additionally we have to explain that the length of the paragraphs results from editorial requirements and guidelines.
Authors used clinical and laboratory results of 2289 patients hospitalized for nephrolithiasis in the highest reference hospital. And 9 were diagnosed with acromegaly. Based on this data alone, the authors concluded that “In the population of patients with urolithiasis compared to the general population, the prevalence of acromegaly was almost 50-fold higher.”
In my opinion, nephrolithiasis patients in highest reference hospital do not represent the whole patients. This is a serious selection bias. Basically, I think this study design is wrong. Therefore, the results of the study are also unreliable.
We agree to the fact that patients hospitalized for nephrolithiasis who took part in the study do not represent the whole population of patients with this disease. However, this subgroup represented patients with urolithiasis who demanded urological interventions described in the text. Thus we corrected the text to detail the message.
Reviewer 2 Report
I really thank the editorial team for inviting me to review this manuscript.
Introduction :
« depends on the supply of fluids and diet, as well as genetic predisposition. » please do not position the genetics and diet at the same level. Diet disorders are clearly more frequent.
“However, the differences in the prevalence of nephrolithiasis in the different populations, due to some degree of unification of eating patterns, will probably decrease in the future[7].” Please remove. Planned prevalence of KS is 30% in 2050 in a recent literature review. Eating habits will not be similar among populations to my opinion.
« URSL (UreteroRenoScopic Lithotripsy) is a method based on crushing stones using an automatic or laser lithotripter after inserting a rigid ureteroscope into the ureter and renal pelvis [9,10]” again this is not true. Automatic does not exist and in renal and ureter cavities, more and more flexible URS are used.
The catalog of Urologic procedures is not really useful. A quick summarization would be enough.
The main and secondary objectives of this study are not clearly defined.
Methods
This section lack of details. Please modify
Results are well presented. We are lacking the stone sizes or dimensions. Please consider to report it. It helps a lot in understanding the repartition of procedures.
Discussion
Is adapted to the topic.
Conclusion
Adequate
Author Response
Dear Reviewer,
First of all, we would kindly like to thank you for the review. We have corrected the manuscript according to all your valuable comments and suggestions. We hope the corrected manuscript will meet your all expectations. Below, we have attached the answers for all of your questions.
Introduction :
« depends on the supply of fluids and diet, as well as genetic predisposition. » please do not position the genetics and diet at the same level. Diet disorders are clearly more frequent.
Paragraph had been updated.
“However, the differences in the prevalence of nephrolithiasis in the different populations, due to some degree of unification of eating patterns, will probably decrease in the future[7].” Please remove. Planned prevalence of KS is 30% in 2050 in a recent literature review. Eating habits will not be similar among populations to my opinion.
Original text removed due to suggestion The paragraph was reconstructed.
« URSL (UreteroRenoScopic Lithotripsy) is a method based on crushing stones using an automatic or laser lithotripter after inserting a rigid ureteroscope into the ureter and renal pelvis [9,10]” again this is not true. Automatic does not exist and in renal and ureter cavities, more and more flexible URS are used.
Paragraph had been updated due to accurate comment. Originally we used an unfortunate arrangement of words that distorted the intended message.
The catalog of Urologic procedures is not really useful. A quick summarization would be enough.
We need to explain that the length of some paragraphs results from editorial requirements (number of words).
The main and secondary objectives of this study are not clearly defined.
Objectives had been revised and updated.
Methods
This section lack of details. Please modify
The section had been updated
Results are well presented.
We are lacking the stone sizes or dimensions. Please consider to report it.
It helps a lot in understanding the repartition of procedures.
Unfortunately all of stone sizes and dimensions had not been archived, thus we were not be able to provide high-quality data.
Round 2
Reviewer 1 Report
As I said in the first review, nephrolithiasis patients in highest reference hospital do not represent the whole patients. This paper has a serious selection bias and this study design is wrong.
But, acromegaly is a rare disease, and papers on its relationship to nephrolithiasis are rare, too.
So I don't think this paper is good, but I think it has meaning in its own way.
Author Response
As I said in the first review, nephrolithiasis patients in highest reference hospital do not represent the whole patients. This paper has a serious selection bias and this study design is wrong. But, acromegaly is a rare disease, and papers on its relationship to nephrolithiasis are rare, too. So I don't think this paper is good, but I think it has meaning in its own way.
Once again, we would like to kindly thank you for reviewing the manuscript and taking the time to familiarize yourself with it. We are aware of the limitations of our work and the imperfect study design; however, we believe that the presented data can shed light on the overlooked problem of the correlation between acromegaly and nephrolithiasis.
Best regards
Authors
Reviewer 2 Report
I thank the editorial board for inviting me to review this manuscript.
Many improvements have been made by authors and I have to congratulate them for this.
Small remark : “L152 The aim of the manuscript was analyze correlation between acromegaly and nephrolithiasis in the group of patients, who required hospitalization in highest reference center (and urological intervention).” Please consider to end the introduction section by this sentence that state the main objective of your manuscript.
Author Response
I thank the editorial board for inviting me to review this manuscript.
Many improvements have been made by authors and I have to congratulate them for this.
Small remark : “L152 The aim of the manuscript was analyze correlation between acromegaly and nephrolithiasis in the group of patients, who required hospitalization in highest reference center (and urological intervention).” Please consider to end the introduction section by this sentence that state the main objective of your manuscript.
Once again, we would like to kindly thank you for reviewing the manuscript and taking the time to familiarize yourself with it. We have corrected the manuscript due to recommendation.
Best regards
Authors